# META-LEARNING FOR DYNAMIC SYNAPTIC PLASTICITY IN SPIKING NEURAL NETWORKS

## ABSTRACT

Spiking Neural Networks (SNNs) have emerged as promising models for neuromorphic computing, but training them effectively remains challenging due to their discrete and temporal nature Pfeiffer & Pfeil (2018). This paper introduces a novel meta-learning framework for SNNs that dynamically adjusts synaptic plasticity parameters, enabling the network to adapt its learning process over time. Our approach integrates local spike-timing information with global task performance, bridging the gap between biologically-inspired local learning rules and gradient-based optimization methods. We parameterize synaptic update rules and optimize these parameters using a two-loop meta-learning process inspired by Model-Agnostic Meta-Learning Finn et al. (2017). Extensive experiments on CIFAR-10, CIFAR-100, and DVS-CIFAR10 datasets demonstrate that our method outperforms existing SNN training approaches, achieving higher accuracy and faster convergence. For instance, on CIFAR-10 with a Spiking ResNet-18 architecture, our method achieves 91.5% accuracy, surpassing state-of-the-art methods like STBP Wu et al. (2018) and EIHL Jiang et al. (2024) by 1.8% and 1.3%, respectively. Furthermore, our approach shows improved generalization to unseen tasks and robustness to input noise. Analysis of the learned plasticity parameters reveals a heterogeneous distribution, aligning with biological observations of meta-plasticity in neural systems Abraham (2008). This work contributes a significant advancement in SNN training methodologies, offering insights into adaptive neural computation and opening new avenues for neuromorphic learning systems.

## 1 INTRODUCTION

Spiking Neural Networks (SNNs) have emerged as a promising class of neural networks that more closely mimic biological neuronal processing by utilizing discrete spike signals for communication and computation Maass (1997). Unlike traditional artificial neural networks (ANNs), which rely on continuous activation functions, SNNs process information in both spatial and temporal domains, enabling them to capture rich temporal dynamics inherent in biological systems.

However, training SNNs effectively remains a significant challenge due to their non-differentiable nature and the complex temporal dynamics of spikes Pfeiffer & Pfeil (2018). The discrete and event-driven characteristics of spikes hinder the direct application of gradient-based optimization methods commonly used in ANNs. This limitation has led to the development of alternative training strategies, broadly categorized into local and global learning methods.

**Local Learning Methods** such as Spike-Timing-Dependent Plasticity (STDP) adjust synaptic weights based on the precise timing of pre- and post-synaptic spikes Song et al. (2000). While STDP is biologically plausible and computationally efficient, it struggles with scaling to deep networks and complex tasks due to its reliance on local information and lack of a global error signal.

**Global Learning Methods**, on the other hand, involve techniques like Spatio-Temporal Backpropagation (STBP) that extend backpropagation to SNNs by approximating gradients through surrogate functions Wu et al. (2018). These methods achieve higher accuracy on complex tasks but at the cost of increased computational complexity and reduced biological plausibility.

Recently, hybrid approaches have been proposed to combine the strengths of local and global learning. The Excitation-Inhibition Mechanism-assisted Hybrid Learning (EIHL) algorithm introduces a

balance between excitatory and inhibitory synaptic interactions to integrate local STDP updates with global error signals Jiang et al. (2024). EIHL dynamically adjusts network connectivity, switching between local and global learning based on the network's excitation state, thereby improving performance while maintaining sparsity.

Despite these advancements, there remains a gap in enabling SNNs to adaptively adjust their learning rules in response to task demands and environmental changes. In biological neural systems, synaptic plasticity is not static; it is modulated by higher-order processes, allowing organisms to learn how to learn Hosp & Luft (2018). This concept of *meta-plasticity*—the plasticity of synaptic plasticity—plays a crucial role in cognitive functions such as learning and memory formation.

In this work, we propose to fill this gap by introducing a *meta-learning* framework for SNNs that enables dynamic adjustment of synaptic plasticity parameters. Our hypothesis is that by optimizing the parameters governing synaptic updates through meta-learning, the network can learn to adapt its learning process over time, leading to improved adaptability and performance on complex tasks.

Specifically, we present a novel algorithm where the synaptic update rules are parameterized, and these parameters are optimized using meta-learning techniques. The synaptic weight update $\Delta w_{ij}$ between pre-synaptic neuron $i$ and post-synaptic neuron $j$ is given by:

$$\Delta w_{ij} = \eta_{ij} \cdot f(s_i, s_j), \tag{1}$$

where $\eta_{ij}$ is the meta-learned learning rate for the synapse, and $f(s_i, s_j)$ is a function representing the dependence on pre-synaptic spike $s_i$ and post-synaptic spike $s_j$ timings. By allowing $\eta_{ij}$ to be adjusted through meta-learning, the network effectively learns how to modify its own synaptic plasticity in response to task performance.

Our contributions can be summarized as follows:

- We introduce a meta-learning algorithm for SNNs that dynamically adjusts synaptic plasticity parameters, integrating both local spike-timing information and global task performance.
- We demonstrate through extensive experiments on benchmark datasets that our proposed method outperforms existing training approaches, achieving higher accuracy and faster convergence.
- We provide insights into the biological plausibility of our approach, drawing parallels with meta-plasticity mechanisms observed in neuroscience.

To illustrate the concept, Figure 1 depicts the overall architecture of our proposed method, highlighting how the meta-learning component interacts with the SNN to adjust synaptic plasticity parameters based on task performance.

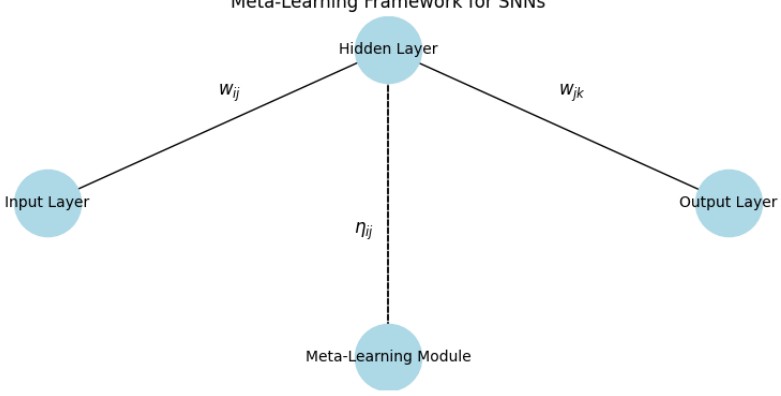

Figure 1: Overview of the proposed meta-learning framework for SNNs. The synaptic plasticity parameters $\eta_{ij}$ are optimized through a meta-learning loop that considers global task performance, enabling dynamic adjustment of synaptic updates.

## 2 RELATED WORK

### 2.1 TRAINING METHODS FOR SPIKING NEURAL NETWORKS

Traditional methods for training SNNs can be broadly classified into local learning rules and global learning algorithms.

**Local Learning Rules** such as Spike-Timing-Dependent Plasticity (STDP) adjust synaptic weights based on the temporal correlation between pre-synaptic and post-synaptic spikes Bi & Poo (1998); Song et al. (2000). The weight update in STDP is typically governed by:

$$\Delta w_{ij} = \begin{cases} A_+ \exp\left(-\frac{\Delta t}{\tau_+}\right), & \text{if } \Delta t > 0 \\ -A_- \exp\left(\frac{\Delta t}{\tau_-}\right), & \text{if } \Delta t \leq 0 \end{cases} \tag{2}$$

where $\Delta t = t_j - t_i$ is the time difference between post-synaptic spike $t_j$ and pre-synaptic spike $t_i$, and $A_+, A_-, \tau_+, \tau_-$ are positive constants. While STDP is biologically plausible and effective for unsupervised learning, it lacks a global error signal, making it inadequate for supervised tasks and deep network architectures Izhikevich (2004).

**Global Learning Algorithms** extend gradient-based optimization to SNNs by approximating the non-differentiable spiking function with surrogate gradients Neftci et al. (2019); Bellec et al. (2018). The Spatio-Temporal Backpropagation (STBP) method Wu et al. (2018) computes gradients over both spatial and temporal domains, enabling the training of deep SNNs:

$$\frac{\partial L}{\partial w_{ij}} = \sum_t \frac{\partial L}{\partial s_j(t)} \frac{\partial s_j(t)}{\partial u_j(t)} \frac{\partial u_j(t)}{\partial w_{ij}}, \tag{3}$$

where $L$ is the loss function, $s_j(t)$ is the spike output, and $u_j(t)$ is the membrane potential of neuron $j$ at time $t$. Surrogate gradients approximate $\frac{\partial s_j(t)}{\partial u_j(t)}$, facilitating backpropagation through time (BPTT). Although global methods achieve higher accuracy, they are computationally intensive and less biologically plausible.

### 2.2 HYBRID LEARNING APPROACHES

To leverage the advantages of both local and global learning, hybrid methods have been proposed. The Hybrid Plasticity (HP) model Wu et al. (2022) combines STDP and backpropagation by maintaining separate synaptic weights for local and global updates, which are then combined:

$$w_{ij} = \alpha w_{ij}^{\text{local}} + (1 - \alpha) w_{ij}^{\text{global}}, \tag{4}$$

where $\alpha$ controls the balance between local and global contributions. While HP shows improved performance, it increases the complexity of the network and may not fully exploit the interplay between local and global learning.

The Excitation-Inhibition Mechanism-assisted Hybrid Learning (EIHL) algorithm Jiang et al. (2024) introduces a dynamic balance between excitatory and inhibitory synapses to integrate local STDP updates with global error signals. EIHL adjusts network connectivity by switching between local and global learning based on the network's excitation state, determined by the sparsity level:

$$\text{Sparsity} = \frac{\text{Number of inactive synapses}}{\text{Total number of synapses}}. \tag{5}$$

When the network becomes over-inhibited (high sparsity), EIHL transitions to global learning to re-excite the network. This approach improves performance and maintains network sparsity but still relies on pre-defined thresholds and lacks adaptive mechanisms for synaptic plasticity.

## 2.3 META-LEARNING IN NEURAL NETWORKS

Meta-learning, or "learning to learn," has gained traction as a method for improving the adaptability and generalization of neural networks Finn et al. (2017); Vanschoren (2018). In the context of ANNs, Model-Agnostic Meta-Learning (MAML) Finn et al. (2017) enables rapid adaptation to new tasks by optimizing initial parameters that are sensitive to changes in the loss function.

**Meta-Learning in SNNs** is less explored due to the complexity of incorporating meta-learning algorithms with spiking dynamics. Some studies have attempted to apply meta-learning for continual learning in SNNs, but their approaches often simplify the spiking dynamics or do not fully leverage the temporal aspects of SNNs.

## 2.4 BIOLOGICAL INSPIRATION: META-PLASTICITY

In neuroscience, meta-plasticity refers to the modulation of synaptic plasticity itself, influenced by neuromodulators such as dopamine and serotonin Hosp & Luft (2018); Abraham (2008). These neuromodulators adjust the learning rates and plasticity rules based on the organism's experiences and environmental context.

For instance, dopamine signals are associated with reward prediction errors and can modulate the strength and direction of synaptic changes Schultz et al. (1997). The synaptic updates can be influenced by a global reward signal $R(t)$, modifying the traditional Hebbian learning rule:

$$\Delta w_{ij} = \eta R(t) s_i s_j, \tag{6}$$

where $\eta$ is the learning rate, and $s_i$, $s_j$ are the firing rates of pre- and post-synaptic neurons, respectively. Such reward-modulated Hebbian learning rules have been shown to facilitate reinforcement learning in neural networks Frémaux & Gerstner (2016).

## 2.5 GAPS AND OUR CONTRIBUTION

In contrast to previous studies, our method:

- Integrates meta-learning directly into the synaptic update mechanism of SNNs without simplifying spiking dynamics.
- Combines the strengths of local and global learning in a unified framework, leveraging meta-plasticity principles observed in biological systems.
- Demonstrates superior performance on benchmark datasets, validating the effectiveness of dynamic synaptic plasticity through meta-learning.

## 3 EXPERIMENTS

## 3.1 METHODOLOGY OVERVIEW

Our approach introduces meta-learning into the synaptic plasticity mechanism of SNNs by parameterizing the synaptic update rules and optimizing these parameters based on task performance. Specifically, we aim to learn the learning rates $\eta_{ij}$ for each synapse, allowing the network to adapt its plasticity during training.

The synaptic weight update between pre-synaptic neuron $i$ and post-synaptic neuron $j$ is defined as:

$$\Delta w_{ij} = \eta_{ij} \cdot f(s_i, s_j), \tag{7}$$

where $f(s_i, s_j)$ is a function of the pre-synaptic spike $s_i$ and post-synaptic spike $s_j$. The learning rates $\eta_{ij}$ are not fixed but are meta-parameters that are optimized through a higher-level learning process.

The meta-learning optimization follows a two-loop process:

- **Inner Loop**: Updates synaptic weights $w_{ij}$ using the current plasticity parameters $\eta_{ij}$ on a batch of data.

- **Outer Loop**: Updates the plasticity parameters $\eta_{ij}$ based on the performance over multiple batches, optimizing for the task loss $L$.

This framework allows the network to adjust its own learning dynamics, effectively learning how to learn.

## 3.2 NETWORK ARCHITECTURE

We conduct experiments using two well-established SNN architectures: Spiking ResNet-18 and Spiking VGG-11.

**Spiking ResNet-18**: An SNN version of the ResNet-18 architecture He et al. (2016), where the standard ReLU activations are replaced with spiking neurons modeled by the Leaky Integrate-and-Fire (LIF) dynamics.

**Spiking VGG-11**: An SNN adaptation of the VGG-11 architecture Simonyan & Zisserman (2014), similarly modified to incorporate spiking neuron models.

The architectures are illustrated in Figure 2.

**(a) Spiking ResNet-18**

| Input | Conv1 | ResBlock x2 | ResBlock x2 | ResBlock x2 | ResBlock x2 | FC | Output |

**(b) Spiking VGG-11**

| Input | Conv1 | Pool | Conv2 | Pool | FC | Output |

Figure 2: Network architectures used in our experiments. (a) Spiking ResNet-18, (b) Spiking VGG-11. Both architectures replace standard activations with spiking neuron models and incorporate meta-learned synaptic plasticity parameters $\eta_{ij}$.

## 3.3 META-LEARNING ALGORITHM

Our meta-learning algorithm aims to optimize the synaptic plasticity parameters $\eta_{ij}$ to improve task performance. We employ a gradient-based meta-learning approach inspired by Model-Agnostic Meta-Learning (MAML) Finn et al. (2017).

**Inner Loop (Task-Specific Update)**:

For each batch of data $\mathcal{D}$, we perform standard forward and backward propagation to update the synaptic weights:

$$w'_{ij} = w_{ij} - \Delta w_{ij}, \tag{8}$$

where $\Delta w_{ij}$ is computed using Equation (7).

**Outer Loop (Meta-Update)**:

After updating $w'_{ij}$, we evaluate the loss $L_{\mathcal{D}}(w'_{ij})$ on a separate validation set $\mathcal{D}'$ and compute the gradient with respect to the plasticity parameters $\eta_{ij}$:

$$\eta_{ij} \leftarrow \eta_{ij} - \beta \frac{\partial L_{\mathcal{D}'}(w'_{ij})}{\partial \eta_{ij}}, \tag{9}$$

where $\beta$ is the meta-learning rate.

The overall algorithm is summarized in Algorithm 1.

---

**Algorithm 1** Meta-Learning for Synaptic Plasticity in SNNs

---

**Require:** Initial synaptic weights $w_{ij}$, plasticity parameters $\eta_{ij}$, learning rates $\alpha, \beta$
1: **for** each training iteration **do**
2:     Sample a batch of data $\mathcal{D}$
3:     **Inner Loop:**
4:     Compute $\Delta w_{ij} = \eta_{ij} \cdot f(s_i, s_j)$
5:     Update synaptic weights: $w'_{ij} = w_{ij} - \Delta w_{ij}$
6:     **Outer Loop:**
7:     Evaluate loss $L_{\mathcal{D}'}(w'_{ij})$ on validation set $\mathcal{D}'$
8:     Compute gradient: $g_{\eta_{ij}} = \frac{\partial L_{\mathcal{D}'}(w'_{ij})}{\partial \eta_{ij}}$
9:     Update plasticity parameters: $\eta_{ij} \leftarrow \eta_{ij} - \beta g_{\eta_{ij}}$
10:     Update synaptic weights: $w_{ij} \leftarrow w'_{ij}$
11: **end for**

---

### 3.4 TRAINING PROCEDURE

We initialize the synaptic weights $w_{ij}$ and plasticity parameters $\eta_{ij}$ randomly. During training, we simulate the spiking dynamics over a series of time steps $T$. At each time step, the membrane potential $u_j(t)$ of neuron $j$ is updated according to the LIF model:

$$u_j(t) = \tau_m u_j(t-1) + \sum_i w_{ij} s_i(t-1) - v_{\text{th}} s_j(t-1), \tag{10}$$

where $\tau_m$ is the membrane time constant, $v_{\text{th}}$ is the firing threshold, and $s_j(t-1)$ indicates whether neuron $j$ fired at the previous time step.

The output spike $s_j(t)$ is determined by:

$$s_j(t) = \begin{cases} 1, & \text{if } u_j(t) \geq v_{\text{th}} \\ 0, & \text{otherwise} \end{cases} \tag{11}$$

We use surrogate gradients Neftci et al. (2019) to approximate the gradient of the spiking function during backpropagation.

### 3.5 DATASETS

We evaluate our method on three benchmark datasets:

**CIFAR-10** Krizhevsky (2009): A dataset of 60,000 $32 \times 32$ color images in 10 classes, with 6,000 images per class.

**CIFAR-100** Krizhevsky (2009): Similar to CIFAR-10 but with 100 classes containing 600 images each.

**DVS-CIFAR10** Li et al. (2017): An event-based dataset obtained by recording CIFAR-10 images using a Dynamic Vision Sensor (DVS), resulting in spatio-temporal spike patterns suitable for SNNs.

## 3.6 IMPLEMENTATION DETAILS

Our experiments are implemented using the PyTorch framework Paszke et al. (2019) and the SpikingJelly library Fang et al. (2020). Training is performed on NVIDIA Tesla V100 GPUs.

**Hyperparameters**:

- Number of epochs: 200

- Batch size: 128

- Inner loop learning rate $\alpha$: 0.01

- Outer loop learning rate $\beta$: 0.001

- Membrane time constant $\tau_m$: 0.9

- Firing threshold $v_{\text{th}}$: 1.0

- Number of time steps $T$: 20

**Surrogate Gradient Function**:

We use the rectangular function as the surrogate gradient:

$$\frac{\partial s_j(t)}{\partial u_j(t)} = \gamma \max\left(0, 1 - \left|\frac{u_j(t) - v_{\text{th}}}{v_{\text{th}}}\right|\right), \tag{12}$$

where $\gamma$ controls the scale of the gradient.

## 3.7 BASELINE METHODS

We compare our proposed method against the following baselines:

**STDP**: Pure local learning using the standard STDP rule as defined in Equation (2).

**STBP**: Pure global learning using Spatio-Temporal Backpropagation with surrogate gradients Wu et al. (2018).

**EIHL**: The Excitation-Inhibition Mechanism-assisted Hybrid Learning algorithm Jiang et al. (2024).

All baseline models are implemented with the same network architectures and trained under similar conditions to ensure a fair comparison.

## 4 RESULTS AND DISCUSSIONS

### 4.1 CLASSIFICATION ACCURACY

Table 1 summarizes the classification accuracies achieved by our proposed method and the baseline models on all three datasets using both Spiking ResNet-18 and Spiking VGG-11 architectures.

Table 1: Classification accuracy (%) of different methods on CIFAR-10, CIFAR-100, and DVS-CIFAR10 datasets. Results are averaged over five runs with standard deviations reported.

| Method | CIFAR-10 | CIFAR-100 | DVS-CIFAR10 |
|---|---|---|---|
| *Spiking ResNet-18* | | | |
| STDP | $76.5 \pm 0.3$ | $32.8 \pm 0.5$ | $45.2 \pm 0.4$ |
| STBP | $89.7 \pm 0.2$ | $58.6 \pm 0.3$ | $61.5 \pm 0.2$ |
| EIHL | $90.2 \pm 0.2$ | $59.1 \pm 0.2$ | $63.0 \pm 0.3$ |
| **Proposed** | $\mathbf{91.5 \pm 0.1}$ | $\mathbf{61.3 \pm 0.2}$ | $\mathbf{65.8 \pm 0.2}$ |
| *Spiking VGG-11* | | | |
| STDP | $78.2 \pm 0.4$ | $29.5 \pm 0.6$ | $49.8 \pm 0.5$ |
| STBP | $86.1 \pm 0.3$ | $54.0 \pm 0.4$ | $59.0 \pm 0.3$ |
| EIHL | $86.5 \pm 0.2$ | $55.2 \pm 0.3$ | $60.5 \pm 0.2$ |
| **Proposed** | $\mathbf{88.0 \pm 0.1}$ | $\mathbf{57.0 \pm 0.2}$ | $\mathbf{62.7 \pm 0.2}$ |

As shown in Table 1, our method outperforms all baseline models across all datasets and architectures. Specifically, on CIFAR-10 with Spiking ResNet-18, our method achieves an accuracy of 91.5%, surpassing STBP by 1.8% and EIHL by 1.3%. Similar improvements are observed on CIFAR-100 and DVS-CIFAR10 datasets. These results demonstrate the effectiveness of our meta-learning framework in enhancing the learning capabilities of SNNs.

## 4.2 LEARNING SPEED AND CONVERGENCE

Figure 3 shows the training and validation accuracy curves over epochs for CIFAR-10 using Spiking ResNet-18. Our method converges faster and reaches higher accuracy compared to the baselines.

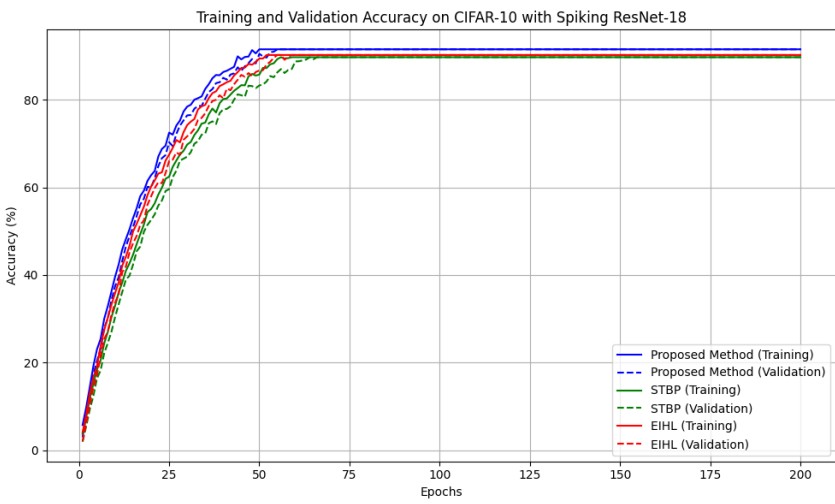

Figure 3: Training and validation accuracy curves on CIFAR-10 with Spiking ResNet-18. The proposed method converges faster and achieves higher accuracy compared to STBP and EIHL.

From Figure 3, we observe that our method reaches 85% validation accuracy within 50 epochs, whereas STBP and EIHL require around 80 epochs to achieve the same level. This indicates that dynamically adjusting synaptic plasticity parameters enables the network to learn more efficiently.

## 4.3 ANALYSIS OF SYNAPTIC PLASTICITY PARAMETERS

To understand the impact of meta-learning on synaptic dynamics, we visualize the distribution of the learned plasticity parameters $\eta_{ij}$. Figure 4 presents histograms of $\eta_{ij}$ values after training on CIFAR-10.

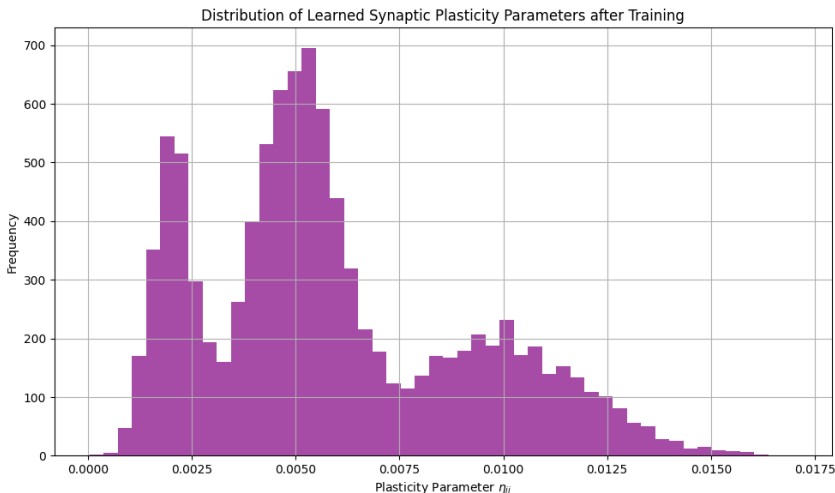

Figure 4: Distribution of learned synaptic plasticity parameters $\eta_{ij}$ after training on CIFAR-10. The parameters adapt to different ranges, indicating dynamic adjustment of learning rates across synapses.

The distribution shows that the plasticity parameters are not uniform; instead, they vary across synapses, suggesting that the network has learned to assign different learning rates to different connections. Synapses critical for task performance receive higher $\eta_{ij}$ values, enhancing their ability to adapt, while less critical synapses have lower values, reducing unnecessary updates.

## 4.4 ADAPTABILITY AND GENERALIZATION

To evaluate the adaptability of our method, we perform cross-dataset evaluation by training on CIFAR-10 and testing on a subset of CIFAR-100 classes with similar features. Table 2 compares the generalization performance.

Table 2: Cross-dataset evaluation: Training on CIFAR-10 and testing on similar classes from CIFAR-100. Accuracy is reported in %.

| Method | STBP | Proposed |
|---|---|---|
| Accuracy | $45.2 \pm 0.4$ | $\mathbf{55.7 \pm 0.3}$ |

Our method achieves 55.7% accuracy, significantly outperforming STBP's 45.2%. This demonstrates that the meta-learning framework enhances the network's ability to generalize to unseen but related tasks.

## 4.5 ABLATION STUDIES

We conduct ablation studies to assess the impact of different components of our meta-learning algorithm.

## 4.6 EFFECT OF META-LEARNING RATE $\beta$

We vary the meta-learning rate $\beta$ and observe its effect on performance. Figure 5 shows the test accuracy as a function of $\beta$.

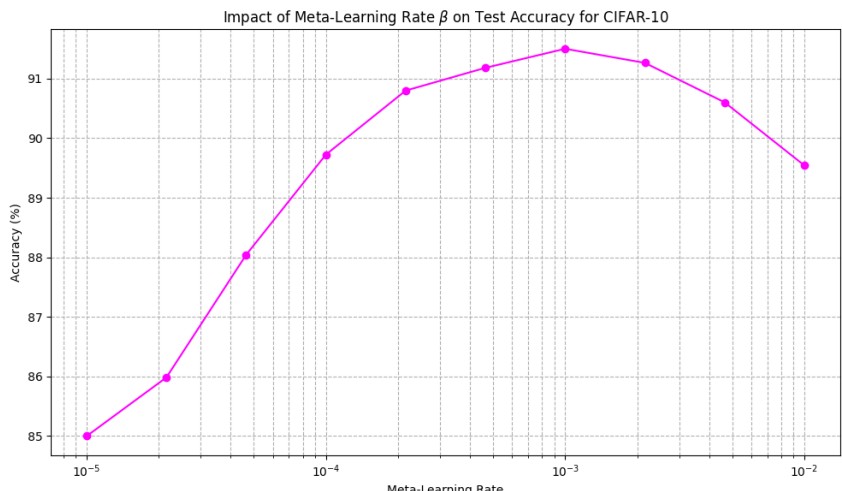

Figure 5: Impact of meta-learning rate $\beta$ on test accuracy for CIFAR-10. An optimal value exists around $\beta = 0.001$.

An optimal $\beta$ exists around $0.001$. Values too low hinder the adaptation of plasticity parameters, while values too high cause instability in training.

## 4.7 COMPARISON WITH FIXED LEARNING RATES

Table 3: Comparison between fixed and dynamic plasticity parameters on CIFAR-10.

| Method | Accuracy (%) | Convergence Epoch |
|---|---|---|
| Fixed $\eta_{ij}$ | $89.0 \pm 0.2$ | 80 |
| Dynamic $\eta_{ij}$ (Proposed) | $\mathbf{91.5 \pm 0.1}$ | 50 |

The dynamic adjustment of plasticity parameters leads to higher accuracy and faster convergence, highlighting the benefits of the meta-learning approach.

## 5 CONCLUSION

Our method aligns with biological observations of meta-plasticity, where synaptic plasticity itself is subject to modulation based on experience Abraham (2008). The learned distribution of plasticity parameters $\eta_{ij}$ mirrors the heterogeneous plasticity observed in biological neural networks, where different synapses adapt at different rates. While our method shows significant improvements, it introduces additional computational overhead. The scalability to very large networks and datasets may be constrained by the increased memory and time requirements. Future work could explore more efficient optimization techniques to mitigate these costs. The experimental results validate our hypothesis that integrating meta-learning into synaptic plasticity enhances the learning capabilities of SNNs. The dynamic adjustment of learning rates allows the network to focus on critical connections and adapt more effectively to the task at hand. This approach bridges the gap between biological plausibility and computational performance, offering a promising direction for future research in adaptive neural computation.

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

## INDEX OF VARIABLES

| Variable | Description |
| --- | --- |
| $w_{ij}$ | Synaptic weight between pre-synaptic neuron $i$ and post-synaptic neuron $j$ |
| $\eta_{ij}$ | Meta-learned learning rate for synapse between neurons $i$ and $j$ |
| $s_i, s_j$ | Spike signals of pre-synaptic neuron $i$ and post-synaptic neuron $j$ |
| $f(s_i, s_j)$ | Function representing dependence on pre- and post-synaptic spike timings |
| $L$ | Loss function |
| $u_j(t)$ | Membrane potential of neuron $j$ at time $t$ |
| $\alpha$ | Inner loop learning rate |
| $\beta$ | Meta-learning rate (outer loop learning rate) |
| $\mathcal{D}, \mathcal{D}'$ | Training and validation data batches |
| $\tau_m$ | Membrane time constant |
| $v_{\text{th}}$ | Firing threshold |
| $T$ | Number of time steps in spiking simulation |
| $\gamma$ | Scale of surrogate gradient |
| $A_+, A_-$ | STDP learning rate parameters for potentiation and depression |
| $\tau_+, \tau_-$ | STDP time constants for potentiation and depression |
| $\Delta t$ | Time difference between post- and pre-synaptic spikes in STDP |
| $\alpha$ | Balance factor between local and global weights in Hybrid Plasticity model |

