# OpenReview forum: "Meta-Learning for Dynamic Synaptic Plasticity in Spiking Neural Networks"
_ICLR.cc/2025/Conference — Submitted to ICLR 2025_

### Official Review · Reviewer_kkJG · 2024-11-01

**Soundness:** 3
**Presentation:** 2
**Contribution:** 1
**Rating:** 3
**Confidence:** 3

**Summary:**

This paper proposes a new training method for spiking neural networks to adaptively tune the learning rate $\eta$ by updating the learning rate using the gradient of the training loss with respect to the learning rate. The paper demonstrates that the proposed method achieves higher accuracy and faster convergence on three data sets (CIFAR-10, CIFAR-100, and DVS-CIFAR10) and two network architectures (Spiking ResNet-18 and Spiking VGG-11).

**Strengths:**

1. The proposed training method is clearly demonstrated via both pseudocodes and explanations. It is intuitive and easy to understand to adaptively tune the learning rate using the gradient of the training loss with respect to the learning rate.
2. Although codes are not provided, there are sufficient details about the algorithm, hyperparameters, and network architectures.

**Weaknesses:**

First, some details of the proposed method are not clearly explained.
1. In Algorithm 1, if I do not miss anything important, the learning rate $\alpha$ is never used. Meanwhile, there are two $\alpha$'s in the table of variables at the end of the paper.
2. The detailed formulation or explanation of the function $f$ is never mentioned, which makes the pseudocode incomplete.
3. Is the number of inner loops equal to that of outer loops? In the current pseudocode, there is an outer loop after each inner loop. Then the names of "inner loop" and "outer loop" become confusing since they are parallel. Meanwhile, the frequent update of $\eta$ makes the time complexity at least twice.

Second, the details of the experiments are unexplained.
1. Authors mentioned STDP, STBP, HP, and EIHL in related works. However, HP is not compared in all experiments, and only STBP is compared in Table 2. Can authors provide explanations for why not compare them?
2. In Table 1, Spiking ResNet-18 performs better than VGG-11 on almost all methods and data sets but STDP on CIFAR-10 and DVS-CIFAR10 are exceptions. Can authors provide some explanations for this phenomenon?
3. In Figure 3, the authors compare the number of epochs needed to converge for different algorithms. However, the time and space complexity of different algorithms are different. Thus, it is unfair to simply compare the number of epochs. It seems that the proposed method has higher complexity in each epoch, and thus may take more time to converge.
4. It seems that section 4.5 (ablation studies) is not completed.

Finally, the presentation of the paper can be further improved, and here are some suggestions.
1. It seems that all citations are textual ones (using latex command citet). It is better to use parenthetical ones when necessary (using latex command citep), such as " but training them effectively remains challenging due to their discrete and temporal nature (Pfeiffer & Pfeil, 2018)", to make the sentence more readable.
2. The logic in line 36 is a little bit confusing: "Unlike traditional artificial neural networks (ANNs), which rely on continuous activation functions, SNNs process information in both spatial and temporal domains, ...". This sentence compares the domains of ANNs and SNNs, which are not determined by the usage of activation functions. Thus, the clause "which rely on continuous activation functions" seems unnecessary here.
3. Many abbreviations are explained in the main context more than once, such as "Spike-Timing-Dependent Plasticity (STDP)", which occurs both in lines 45 and 115.
4. There are many different models of spiking neural networks, and it is better to introduce the model used in this paper before delving into details.
5. In line 169, it is better to provide some references to "Some studies have attempted to apply meta-learning for continual learning in SNNs" to make the related work more complete.
6. In Figure 2, the authors provide details about network architectures but do not mention the kernel size, stride, padding, and width. Although these parameters are classical ones, it would be better to provide them here since there is plenty of space.
7. In line 317, it is better to provide detailed information about the surrogate gradients. I think it is suitable to move Eq. (12) from line 351 to here.

**Questions:**

See weakness, mainly the first and second ones about algorithm and experiments, respectively.

---

### Official Review · Reviewer_fXU2 · 2024-11-01

**Soundness:** 2
**Presentation:** 3
**Contribution:** 2
**Rating:** 5
**Confidence:** 3

**Summary:**

This submission addresses the important challenge of efficiently training SSNs and introduces a meta-learning framework that dynamically adjusts synaptic plasticity parameters. The authors parameterize the synaptic update rules and optimize the learning rate parameter $\eta_{ij}$ for each $\Delta w_{ij}$ using a two-loop process. They demonstrate that this approach improves upon the recent EIHL method, which relies on pre-defined thresholds and lacks adaptive mechanisms (Jiang et al., ICLR 2024), on CIFAR-10, CIFAR-100, and DVS-CIFAR-10 datasets.

**Strengths:**

•	The problem is well-motivated.

•	The authors tested the approach using well-known standard datasets, showing both training and validation accuracies, and utilized standard SNN architectures.

•	Provided some insights into why the new framework performs well, such as the importance of adaptability (Table 3).

**Weaknesses:**

•	My main concern is the limited evidence demonstrating the technical soundness of this work. For instance, many critical technical details are missing. How were hyperparameters (e.g., network size, batch size) tuned for each model? Without this information, it's unclear if the benchmark models were given the best chance to succeed. Additionally, I could not find the code to reproduce the core results.

•	Very few controls were provided (only the impact of $\beta$ was shown).

•	Insufficient results for certain claims. For example, the benchmark EIHL was not included in Table 2 for examining adaptability and generalization. Furthermore, Section 4.5 on ablation studies did not reference any tables or figures. This and the previous point makes the manuscript seem rushed.

•	Minor typos: citations in lines 076-079 and line 086 should use \citep instead of \citet; line 129 has two periods at the end of the sentence.

•	It appears the performance depends too heavily on tuning $\beta$ (Figure 5); without precise tuning, the performance drops below that of EIHL. This puts the efficiency of this method into question.

•	While this did not impact the score, the discussion of related works could be more comprehensive (e.g., papers by Tim Vogels). The number of cited papers was much lower than what is typical for NeurIPS, ICLR, or ICML submissions.

**Questions:**

•	How does the parameter count compare to the benchmarks?

•	I couldn’t find details on the exact form of $f(s_i,s_j)$; could you clarify what was used?

•	Could you provide more technical details? Specifically, how were the hyperparameters (e.g., network size, batch size) tuned, and how was the train-test split conducted?

---

### Official Review · Reviewer_qkLh · 2024-11-03

**Soundness:** 3
**Presentation:** 3
**Contribution:** 2
**Rating:** 3
**Confidence:** 3

**Summary:**

This paper proposes a meta-learning framework for dynamically adjusting per-synapse learning rates in Spiking Neural Networks (SNNs). The authors claim their method integrates local spike-timing information with global task performance to improve SNN training. They evaluate their approach on image classification tasks using CIFAR-10, CIFAR-100, and DVS-CIFAR10 datasets, comparing against baseline methods including spike-timing dependent plasticity (STDP), spatio-temporal backprop (STBP), and excitation-inhibition hybrid learning (EIHL). The paper reports improved accuracy and faster convergence compared to existing approaches.

**Strengths:**

1. The authors provide a clear motivation for their work, drawing inspiration from biological meta-plasticity.
2. The experimental setup includes comparisons against relevant baseline methods on standard datasets.
3. The general concept of using local learning rules to define the weight update, and global learning rules to set the learning _rate_, is interesting.

**Weaknesses:**

1. **Lack of novelty specific to SNNs**: The core idea of using meta-learning to optimize learning rates is not fundamentally new or specific to SNNs. The authors apply MAML to tune per-synapse learning rates, but this approach could be applied to any neural network architecture. The paper fails to demonstrate how this method uniquely leverages or addresses the specific characteristics of spiking neural networks.
2. **Insufficient comparison to existing optimization methods**: The paper does not compare their approach to well-established methods for setting per-parameter learning rates, such as adaptive optimizers like RMSProp or Adam.
3. **Incomplete ablation studies**: The ablation section (4.5) appears to be unfinished? What ablation studies were conducted, and what did they show?
4. **Unclear what the weight update $f(s_i, s_j)$ is**: How was the function $f$ in equation (7) chosen? I didn't see it defined in the paper.
5. **Limited analysis of the resulting learning rates**: Figure 4 shows a distribution of the final learning rates. How should I interpret this histogram? It looks like there are multiple modes, do those correspond to different layers in the network? Is this distribution sensitive to the network that was trained? What about the random seed?

**Questions:**

- What is the function $f(s_i, s_j)$ in equation (7)? I didn't see a definition of that anywhere.
- From my understanding of the paper, couldn't you apply your method on top of the baseline methods (STDP, STBP, or EIHL), as your method is a way to tune the per-parameter learning rate (with a slight generalization of equation 7)? That is, any optimization method is a function that computes a weight update for weight $w_{ij}$, and couldn't you apply MAML to set the per-parameter learning rate?
- How should I interpret Figure 4?

---

### Official Review · Reviewer_sekH · 2024-11-03

**Soundness:** 1
**Presentation:** 2
**Contribution:** 2
**Rating:** 5
**Confidence:** 4

**Summary:**

This paper proposes an online weight update adaptation algorithm for spiking neural networks that leads to better performance on object recognition tasks. The local weight adaptations driven by STDP are modulated by global signals that are explicitly sensitive to the categorisation loss.

**Strengths:**

Originality
- Inspired by MAML, the authors introduce a two-level training approach for SNNs that is distinct from previous SNN training approaches. This approach provides performance gains and faster training in object recognition tasks.

Significance
- Being able to figure out expressive training regimes for SNNs is critical to enable neuromorphic computing. As such, this approach adds to the possibilities.

**Weaknesses:**

I have one core issues:

Authors state that their algorithm is a meta-learning algorithm. However, it is starkly different from MAML. In MAML, the objective is to train a network such that it can adapt quickly to new tasks. As such, the outer loop gains access to "second-order" information about the possible future task requirements from the inner loop and adapts to make the network adaptable. In this case, the "outer loop" modulates the weight change but also changes the weights directly! In effect, it is unclear if due to the eta updates the network is becoming more adaptable as MAML would suggest. Instead what seems to be happening here is that the eta updates rely on a distinct batch than the weight updates, thereby introducing some generalisable properties in the overall weight updates - seems different from meta-learning. I'd liken this approach more to contextual learning / topdown attentional modulation of the kind that has been proposed in recent works (Linday & Miller, 2018; Cheung et al. 2019; Thorat et al. 2019; Hummos, 2023). Given that this method doesn't exactly resemble MAML, it also makes the "local learning" assumption murky as eta inference requires backpropagation!

Minor critique: Please interpret your results when you present them in the various sections.

Refs:
Lindsay, G. W., & Miller, K. D. (2018). How biological attention mechanisms improve task performance in a large-scale visual system model. ELife, 7, e38105.
Cheung, B., Terekhov, A., Chen, Y., Agrawal, P., & Olshausen, B. (2019). Superposition of many models into one. Advances in neural information processing systems, 32.
Thorat, S., Aldegheri, G., Van Gerven, M. A., & Peelen, M. V . (2019). Modulation of early visual processing
alleviates capacity limits in solving multiple tasks. In 2019 Conference on Cognitive Computational Neuroscience.
Hummos, A (2023). Thalamus: a brain-inspired algorithm for biologically-plausible continual learning and disentangled representations. In The Eleventh International Conference on Learning Representations.

**Questions:**

1. How expensive is this SNN training? Could you compare the FLOPs with a vanilla Resnet/VGG trained with backpropagation? This would position the paper's contribution beyond the field of SNNs
2. If you finish training and set the weights to random init and maintain the trained etas, would the training be faster compared to the case where etas are equal to 1? Ofc you'd have to hyperparameter tune - is the fastest learning possible with eta=1 slower than the learning with the learned etas? Is the best performance better? If yes, you are back in MAML territory and *that* would be a meta-learning setup where you could liken the distribution of etas as something evolution or some other developmental process could figure out.
3. In 2.3 you mention "some studies have attempted to apply meta-learning for CL..." - could you cite them for completeness?
4. What is the nature of the readout? Are those also spiking neurons? Or are you using a softmax layer? (+crossentropy loss on it)
5. How do we interpret the learned eta distribution?
6. In 4.4, how do you perform finetuning? You include a new readout and retrain with your algorithm or just let the weights be updated while keeping etas constant?
7. In 4.7, what does "fixed eta" imply? eta = 1? How is this condition different from STDP? The performance is higher than that of STDP - what is the reason for that?
8. How good is the STDP baseline? How expressive is that learning given that there's no global signal? If you maintain a random Resnet/VGG and just train a readout how much performance do you get?

---

### Official Review · Reviewer_D1qW · 2024-11-04

**Soundness:** 2
**Presentation:** 1
**Contribution:** 2
**Rating:** 5
**Confidence:** 5

**Summary:**

The paper introduces a novel meta-learning approach for training Spiking Neural Networks (SNNs) that adapts per-parameter learning rates using an outer meta-optimization loop during training. The weights are updated with a local plasticity rule in the inner optimization step using a learning rate that is optimized through gradient descent in the outer optimization step. The authors show strong improvements in classification performance on three datasets - CIFAR10, CIFAR100, and DVS-CIFAR10. They also point out interesting links to synaptic metaplasticity in neurobiology.

**Strengths:**

- I found some implications of the proposed method quite interesting. The proposed method still effectively does gradient descent with respect to the loss, except the roles of learning rate and parameters is now partially swapped. It would be interesting to see the exact gradients that are being followed, and the paper would be improved by including exact expressions for the learning dynamics.
- The demonstrated performance improvements are large, and point to the effectiveness of the approach.

**Weaknesses:**

That said, I have several serious concerns about the work. I elaborate below.
- Generally, I found the paper incomplete and not well-written. A lot of important information seems to be missing. For example, the plasticity rule is said to be a function $\delta w_{ij} = \eta_{ij}f(x_i)f(x_j)$, but the function $f$ is never specified. Furthermore, the proposed method is said to be inspired by the MAML approach, but the link to this method is unclear to me. MAML was proposed for multi-task learning and optimized the parameters directly unlike the current method. An introduction to MAML in the prior work section, and a detailed treatment of the actual gradients followed for optimizing the learning rates and how this builds on MAML are important to include.
- On a similar note, how $\frac{\partial L_{\mathcal{D}'}(w'_{ij})}{\partial \eta_{ij}}$ is a quantity that even makes sense is underspecified. I imagine the computation graph includes the plasticity update Eq.8 and is differentiated with autograd in the outer loop with a backward pass, but the text does not provide this information, so I cannot be sure.
- Line 462 says "Synapses critical for task performance receive higher ηij values, enhancing their ability to adapt, while less critical synapses have lower values, reducing unnecessary updates," but wouldn't the opposite be expected? If synapses are critical to performance, shouldn't they become fixed by the end of training? There are interesting links to continual learning approaches here (see [1]) that are not explored. Overall the link to metaplasticity seems thin and should be elaborated more.
- Minor: An overall lack of polish. For example, Fig.1 is very low-resolution and pixelated, and Fig.2 about the network architecture is uninformative and takes up too much space. An inner loop learning rate $\alpha$ is mentioned in line 340 but has not been mentioned otherwise.

[1] Zenke, Friedemann, Ben Poole, and Surya Ganguli. "Continual learning through synaptic intelligence." International conference on machine learning. PMLR, 2017.

**Questions:**

- Papers on SNN typically use spiking datasets such as SHD and SSC to demonstrate their effectiveness. Would the proposed approach also provide a performance improvement there?
- The baselines chosen for comparison seem insufficient for making a strong claim about the metaplasticity approach. How does the proposed method compare against optimizers like Adam which also adapt per-parameter learning rates? It is not mentioned what optimizer is used, so it is difficult to judge the effectiveness of the proposed approach.

---

### Meta-Review · Area_Chair_E3Ga · 2024-12-06

**Metareview:**

The submission lacks technical clarity, with key details about the method missing or poorly explained. Reviewers also found the novelty insufficient, as the proposed approach is not uniquely tailored to spiking neural networks and diverges from standard meta-learning frameworks like MAML. Additionally, incomplete experiments, missing key comparisons, and unclear presentation further weaken the paper. The lack of an author response left critical concerns unaddressed, reinforcing the decision to reject.

**Additional Comments On Reviewer Discussion:**

No author response

---

### Decision · Program_Chairs · 2025-01-22

Reject